# Investigation of Enzyme Immobilization and Clogging Mechanisms in the Enzymatic Synthesis of Amoxicillin

**DOI:** 10.3390/ijms25168557

**Published:** 2024-08-06

**Authors:** Chenyan Fan, Jiale Li, Ruimeng Dong, Yiling Xu, Liqiang Liu

**Affiliations:** College of Food and Biology, Hebei University of Science and Technology, Shijiazhuang 050018, China; 2023111071@stu.hebust.edu.cn (C.F.); 2022111079@stu.hebust.edu.cn (J.L.); 2203020202@stu.hebust.edu.cn (R.D.); 2203020229@stu.hebust.edu.cn (Y.X.)

**Keywords:** enzymatic method, amoxicillin, immobilized enzyme, a blockage, cleanse

## Abstract

This study investigated the blocking mechanism of immobilized penicillin G acylase (PGA) during the enzymatic synthesis of amoxicillin. Laboratory observations revealed that the primary cause of clogging was the crystallization of the substrate and product on the enzyme surface. Adjusting key parameters can significantly reduce clogging and improve catalytic efficiency. Methanol can decrease enzyme activity, but isopropyl alcohol cleaners can effectively remove clogs and protect enzyme activity. These findings provide an experimental foundation for optimizing the PGA immobilization process, which is crucial for achieving high efficiency and sustainability in industrial production.

## 1. Introduction

Due to their great efficacy and broad antibacterial range, β-lactam antibiotics are widely favored as essential antibacterial medications for clinical treatment [1]. However, due to environmental contamination and high expenses, classical chemical synthesis processes are gradually becoming constrained as public awareness of environmental protection and the ongoing quest for sustainable development grow. The enzymatic synthesis of β-lactam medications, particularly the amoxicillin-represented second-generation penicillin antibiotics, has demonstrated numerous benefits [2,3,4].

With its high selectivity and mild reaction conditions, enzymatic production technology significantly reduces the generation of harmful by products. As a result, the conversion rate for the industrial production of amoxicillin has reached more than 99%, aligning with the environmental protection principles of green chemistry [5,6].

Immobilized penicillin G acylase (PGA) is instrumental in the enzymatic synthesis of amoxicillin, primarily due to its significant catalytic role. The immobilization of enzymes enhances their stability and reusability while also simplifying the recovery and purification processes, as noted in the literature. However, the synthesis of amoxicillin via enzymatic methods is a multifaceted procedure governed by intricate dynamics. Factors such as enzyme activity, reaction system temperature, pH value, and substrate concentration interrelate, collectively dictating both the efficiency of the reaction and the quality of the final product [7,8,9].

Acrylic resins are commonly chosen as carrier materials for synthesizing amoxicillin in industrial processes due to their superior chemical and physical characteristics, including outstanding mechanical strength and resistance to microbial degradation [10]. The synthesis of amoxicillin typically involves the utilization of immobilized penicillin G acylase (PGA) within a suspension–suspension system. In this system, the product and substrate are present in the form of insoluble particles or microcrystals, while the enzyme-catalyzed reaction occurs within the aqueous phase. Although this system enhances reaction efficiency to a certain degree, it presents several issues during practical production [11,12]. Particularly, when the activity of immobilized penicillin G acylase (PGA) is excessively high, the synthesis reaction may come to a halt. This phenomenon is likely due to the system’s high viscosity, which leads to the rapid formation of amoxicillin. Consequently, the inconsistency in diffusion rates inside and outside the carrier hinders the free movement of substrate molecules and products into the reaction system. This discrepancy can result in crystallization on the carrier surface or within its void channels, thereby causing blockages [13,14,15,16].

Despite the recognition of this phenomenon, the extent of in-depth research and comprehensive reporting is still insufficient. The present study thoroughly investigates the underlying causes of PGA issues derived from *Achromobacter* sp. and proposes corresponding solution strategies, thereby extensively addressing the existing gaps in research. The integration of these strategies can effectively resolve the blockage issues encountered in the enzymatic synthesis process of β-lactam antibiotics. Additionally, it can significantly enhance both production efficiency and product quality. The findings from this research offer robust scientific support for advancing the realization of green, efficient, and sustainable drug production.

## 2. Results and Discussion

### 2.1. Exploring the Causes of Blockage

This study meticulously examined the plugging phenomenon using scanning electron microscopy (SEM). SEM images revealed that the surfaces of blocked immobilized PGA were densely covered with white crystalline substances. These substances were preliminarily identified as accumulations of substrates and products (Figure 1a–d). In contrast, the surfaces of immobilized PGA without plugging showed no evidence of such crystallization. This contrast underscores a significant association between crystallization formation and enzymatic catalytic activity.

To further investigate, blocked immobilized PGA was cleaned with various solvents, and the resulting solutions were analyzed using high-performance liquid chromatography (HPLC). The results indicated that in 0.2 M phosphate-buffered saline (PBS) at pH 8.0, as well as in 5% methanol and 5% isopropyl alcohol solutions, the proportions of amoxicillin products were 79.29%, 75.46%, and 79.37%, respectively. Other components were present in relatively minor quantities (Figure 2). These findings indicate that the blocking material predominantly consisted of amoxicillin, suggesting its accumulation on the carrier surface and within channels contributed to the premature termination of the reaction.

The results indicate that clogging was related to the suspension–suspension system used in the enzymatic synthesis process. In this process, the highly active immobilized penicillin G acylase (PGA) rapidly catalyzes amoxicillin synthesis, producing a large amount of amoxicillin locally to form a supersaturated solution [17]. Due to its low solubility, amoxicillin quickly crystallizes on the carrier surface, forming a coating. This crystallization leads to blockage, hindering the continuation of the enzyme-catalyzed reaction.

### 2.2. Discussion of Key Influencing Factors in Immobilized Enzyme Blockage

Detailed analysis of substrate concentration, enzyme dosage, rotational speed, and temperature revealed their significant impact on PGA plugging. The experimental results indicate that these key parameters substantially regulate the activity and catalytic efficiency of immobilized PGA.

Precise regulation of substrate concentration is crucial for maintaining the activity of immobilized PGA. Our data indicated that increasing substrate concentration led to an approximately 50% decrease in 6-APA conversion when using high-activity immobilized PGA, demonstrating that non-uniform substrate accumulation can cause blockages (Table 1 (a–e)). Thus, strict control of substrate concentration is imperative for minimizing blockage risk and enhancing catalytic efficiency and product yield [18] (Figure 3A). Notably, altering the amount of injected enzyme has minimal impact on clogging when total enzyme activity remains constant (Table 1 (e, f–h)). This observation highlights the necessity of balancing unit enzyme activity and total enzyme activity in process design (Figure 3B).

Our research provides a different perspective on the impact of rotational speed on the catalytic synthesis of amoxicillin using immobilized PGA. Previous studies suggested that increasing rotational speed can lead to an uneven substance distribution in the reaction system, reducing effective contact between the substrate and enzyme, thus compromising catalytic efficiency [19]. Contrary to these expectations, our observations indicate that the catalytic activity of immobilized PGA and amoxicillin yield remained consistent even when the speed was increased from 600 rpm to 1000 rpm (Table 1 (e, i, j)). This suggests that the highly active immobilized PGA system used in our study is less susceptible to the influence of rotational speed on reaction efficiency than previously assumed (Figure 3C). This novel finding has significant implications for optimizing reaction conditions in industrial settings, suggesting that a broader range of operating speeds can be viable for catalyzing reactions with highly active immobilized enzymes.

Temperature plays a crucial role in immobilized PGA performance. Operating at lower temperatures ensures effective blockage prevention, even when using immobilized PGA with slightly above-normal enzyme activity, namely, over 166.47 U/g. Consequently, 6-APA conversion rates can reach as high as 99% (Table 1 (e, k–n)). This high conversion rate is primarily due to the equilibrium between the external mass transfer rate and the internal reaction rate within the immobilized PGA carrier particles. This balance minimizes substrate and product accumulation on the carrier surface, reducing clogging risk. Additionally, lower temperatures sustain the dynamic balance of the reaction inside immobilized PGA carrier particles, further decreasing blockage and facilitating product release [20,21] (Figure 3D).

This study highlights the critical importance of controlling substrate concentration, enzyme dosage, rotational speed, and temperature during the catalytic synthesis of amoxicillin using immobilized penicillin G Acylase (PGA). The results demonstrate that the meticulous optimization of these parameters is crucial for establishing optimal conditions. Thorough analysis of these key factors can significantly enhance synthesis efficiency and mitigate blockage risk. Consequently, these improvements contribute to the continuous optimization and sustainable development of the amoxicillin production process.

### 2.3. Clogging Enzyme Treatment and Activity Recovery

In this study, we thoroughly evaluated immobilized PGA activity after conducting cleaning with different solvents to investigate the specific effects on enzyme activity (Figure 4). A significant finding was that immobilized PGA activity decreased substantially to 127.71 U/g when an aqueous methanol solution was used for cleaning (Table 2). This reduction can be attributed to methanol’s dehydration effect or its denaturing impact on the enzyme’s protein structure, compromising both the three-dimensional configuration and catalytic efficiency of the enzyme. When isopropyl alcohol was used as a cleaning agent, immobilized PGA activity increased to 173.8 U/g, approximately 10% higher than the enzyme’s initial activity before cleaning, as indicated in Table 2. This indicates that isopropyl alcohol not only removed clogging substances but also minimized adverse impacts on the enzyme’s active center. Consequently, the enzyme’s catalytic activity was maintained or even improved. Moreover, cleaning with alkaline PBS buffer had minimal impact on immobilized PGA activity, maintaining it at 159.56 U/g (Table 2). These findings underscore the importance of selecting an appropriate cleaning solvent for preserving the activity of immobilized PGA. In particular, the exceptional performance of isopropyl alcohol suggests a potential avenue for optimizing the process of cleaning immobilized PGA, enhancing enzyme recycling efficiency and reducing production costs.

## 3. Materials and Methods

### 3.1. Instruments

JJ-1 Precision Timing Electric Mixer (Changzhou Ronghua Instrument Manufacturing Co., Ltd., Changzhou, China), ST2100 Laboratory pH Meter (Ohaus Instrument (Changzhou) Co., Ltd., Changzhou, China), DK-98-II Electric Thermostatic Water Bath (Tianjin Test Instrument Co., Ltd., Tianjin, China), AL104 Electronic Balance (Mettler Toledo Instrument (Shanghai) Co., Ltd., Shanghai, China), 99-1 High-Power Magnetic Heating Agitator (Jintan Ronghua Instrument Manufacturing Co., Ltd., Changzhou, China), Hitachi S-4800 High-Resolution Cold Field Emission Scanning Electron Microscope, and 1260 Infinity LC System (Agilent Technologies, Santa Clara, CA, USA).

### 3.2. Materials

6-Aminopenicillanic acid (6-APA)(Shanghai McLean Biochemical Technology Co., Ltd., Shanghai, China), D-p-hydroxyphenylglycine methyl ester (D-HPGM)(Shanghai McLean Biochemical Technology Co., Ltd., Shanghai, China), LX-1000HA carrier (Xi’an Lanxiao Technology New Material Co., Ltd., Xi’an, China), Anhydrous ethanol (Tianjin Best Chemical Co., Ltd., Tianjin, China), Isopropyl alcohol (Tianjin Best Chemical Co., Ltd., Tianjin, China), Potassium dihydrogen phosphate (Beijing Solaibao Technology Co., Ltd., Beijing, China), Dipotassium hydrogen phosphate (Beijing Solaibao Technology Co., Ltd., Beijing, China), Sodium hydroxide (Beijing Solaibao Technology Co., Ltd., Beijing, China).

### 3.3. Methods

#### 3.3.1. Preparation of Immobilized PGA

##### Carrier Activation

The solution preparation step started with adding 8 mL of glutaraldehyde (50%) and 0.716 g of potassium hydrogen phosphate (K_2_HPO_4_ · 3H_2_O) to deionized water, diluting the mixture to a final volume of 100 mL. After complete dissolution, the solution’s pH was adjusted to 7.8–8.2 using 40% phosphoric acid.

Subsequently, 25 g of the LX-1000HA carrier was added to the solution. The mixture was maintained at 25 °C while stirring at 150 rpm for 1 h. After activation, the carrier was filtered, rinsed with deionized water, drained, and stored for further use.

##### PGA Liquid Enzyme Immobilization

A 0.4 M phosphate buffer solution with a pH range of 6.8–7.2 was prepared. Subsequently, the enzyme solution was prepared using the β-lactam antibiotic synthetase method [22,23]. Various quantities of the activated carrier LX-1000HA were added and immobilized at 25 °C and 150 rpm for 24 h. After immobilization, the enzyme was collected via vacuum filtration, washed with deionized water at twice the volume 3–5 times, and finally filtered and collected for further experimentation.

#### 3.3.2. Determination of Immobilized PGA Activity

##### Definition of Immobilized Enzyme Activity Unit

One unit (1 U) of immobilized enzyme activity is defined as the amount of enzyme required to convert 1 μmol of 6-APA to amoxicillin per gram of enzyme per minute under specified assay conditions [24].

##### Assay Method

Column: YMC-Pack ODS-AQ (250 mm × 4.6 mm, 5 μm). The mobile phase consisted of a 0.05 mol/L potassium dihydrogen phosphate solution, adjusted to pH 5.0 using 2 mol/L of potassium hydroxide, mixed with acetonitrile in a ratio of 97.5:2.5. Detection was performed at a wavelength of 254 nm. The column temperature was maintained at 25 °C, with a flow rate of 1.0 mL/min. The sample injection volume was 20 μL [25].

Reference Solution: Weigh out approximately 25 mg of the amoxicillin reference product and transfer it into a 50 mL volumetric flask. Dissolve the amoxicillin in the mobile phase and dilute it to the mark. Shake well to ensure thorough mixing. Inject 20 μL of this solution into the liquid chromatograph. The resulting chromatogram should match the standard chromatogram.

Sample Solution: Pipette 250 μL of the reaction solution into a 25 mL volumetric flask. Dilute the solution to the mark with the mobile phase and shake well to ensure thorough mixing. Inject 20 μL of the prepared solution into the liquid chromatograph for analysis.

##### Calculation of Synthetase Activity

The formula for calculating the concentration of amoxicillin is
(1)$$C(\mathrm{mmol}/L)=\frac{A_{X}\times W_{R}\times P\times 1000}{A_{R}\times M\times 2.5}$$
where

C is the concentration of amoxicillin in the solution, given in mmol/L;

A_X_ is the sample peak area;

A_R_ is the average peak area of the control substance;

W_R_ is the control quality, given in mg;

M is the relative molecular weight of anhydrous amoxicillin (365.46 g/mol);

P is the control substance content (%).

The formula for calculating enzyme synthesis activity is
(2)$$ASU_{\left(wet\right)}=\frac{C\times V\times 1000}{t\times W}$$
where

ASU represents enzyme synthesis activity, measured in μmol/g·min;

V is the volume of the reaction liquid, given in liters (L);

C is the amoxicillin concentration in the reaction solution, measured in mmol/L;

t is the response time in minutes (min);

W is the mass of the enzyme sample (wet), measured in grams (g).

### 3.4. Causes of Blockage in the Synthesis of Immobilized PGA

#### 3.4.1. Blockage Cause Detection

##### Detection Using an Electron Microscope

To detect causes of blockage in the synthesis of immobilized penicillin G acylase (PGA), small amounts of immobilized PGA samples were dried at 60 °C for 4 h before undergoing conductive treatment in a vacuum evaporator, where a layer of metal film was sprayed onto them. The treated samples were then placed on a conductive slide and transferred to the sample chamber. The chamber was vacuumed to the necessary level, and the acceleration voltage and beam were adjusted to the optimal working range for observing the sample surface [26,27].

#### 3.4.2. Research on Key Parameters

In a 50 mL reaction system, the effects of varying substrate concentrations (6-APA, D-HPGM molar ratio 1:1), enzyme dosages, rotational speed, and temperatures on the synthesis of highly active immobilized PGA were observed (Table 1).

#### 3.4.3. Cleaning Treatment for Clogged Enzymes

The recovered immobilized PGA (1 g) was separately added to 20 mL of pH 8.0 0.2 M PBS buffer, 5% methanol aqueous solution, and 5% isopropanol aqueous solution. The mixtures were incubated and stirred at room temperature for 30 min, followed by washing with three volumes of distilled water. The washing solutions and the cleaned immobilized PGA were subsequently recovered separately. The composition of the cleaning liquid was analyzed using high-performance liquid chromatography (HPLC) and electron microscopy. Subsequently, the synthetic activity of the cleaned immobilized PGA was determined according to the method described in Section 3.3.

### 3.5. Statistical Analysis Technique

IBM SPSS Statistics 27 was utilized to assess the homogeneity of variance using several methods, including the Least Significant Difference (LSD), Duncan’s Multiple Range Test (DMRT), and the Waller–Duncan method (Welch–Duncan).

## 4. Conclusions

In this study, the clogging phenomenon of immobilized PGA in the amoxicillin synthesis process was thoroughly analyzed. Key influencing factors such as substrate concentration, enzyme dosage, rotational speed, and reaction temperature were systematically examined. In this investigation, we utilized scanning electron microscopy (SEM) and high-performance liquid chromatography (HPLC) to elucidate the direct correlation between the clogging phenomenon and the presence of crystalline particles on the enzyme surface.

In industrial applications of immobilized enzyme technology, effective control of the plugging phenomenon is crucial for success. Optimizing reaction conditions, particularly at lower temperatures, enables the immobilization of PGA with a synthetase activity of approximately 170 U/g to effectively mitigate the risk of clogging. This approach has already been implemented in industrial production. Additionally, the use of isopropyl alcohol as a cleaning agent is essential for enhancing the reusability of immobilized enzymes and reducing operational costs.

Our study not only introduces a novel strategy for cleaning and recovering the activity of immobilized enzymes but also establishes a significant scientific basis for regulating enzyme activity in industrial settings. Through the application of appropriate cleaning solvents and the meticulous control of reaction conditions, we can markedly enhance the operational efficiency of immobilized enzymes in industrial synthesis and reduce associated costs.

## Figures and Tables

**Figure 1 ijms-25-08557-f001:**
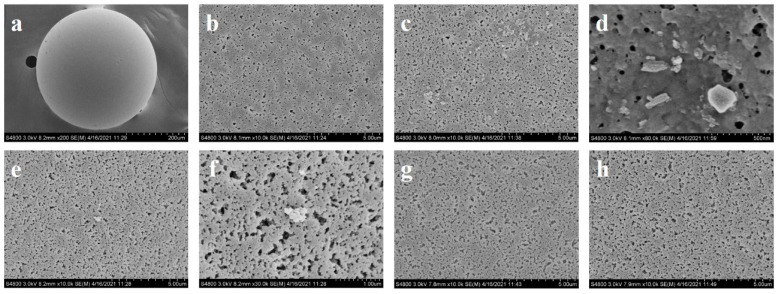
Electron microscope scanning. (**a**) Complete map of normal enzymes; (**b**) local map after normal enzyme activity reaction; (**c**) local map after plugging immobilized PGA reaction; (**d**) local magnification image after plugging immobilized PGA reaction; (**e**) local map of blocked and immobilized PGA after cleaning with PBS buffer was conducted; (**f**) local magnification of blocked and immobilized PGA after cleaning with PBS buffer was conducted; (**g**) local diagram of blocked and immobilized PGA after cleaning with 5% methanol was conducted; (**h**) local map of blocked immobilized PGA after cleaning with 5% isopropyl alcohol was conducted.

**Figure 2 ijms-25-08557-f002:**
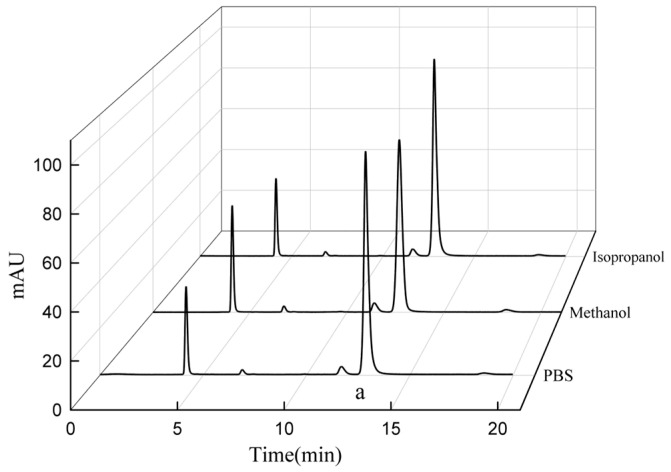
Test results for cleaning using different solvents. Peak a represents amoxicillin in the chromatogram.

**Figure 3 ijms-25-08557-f003:**
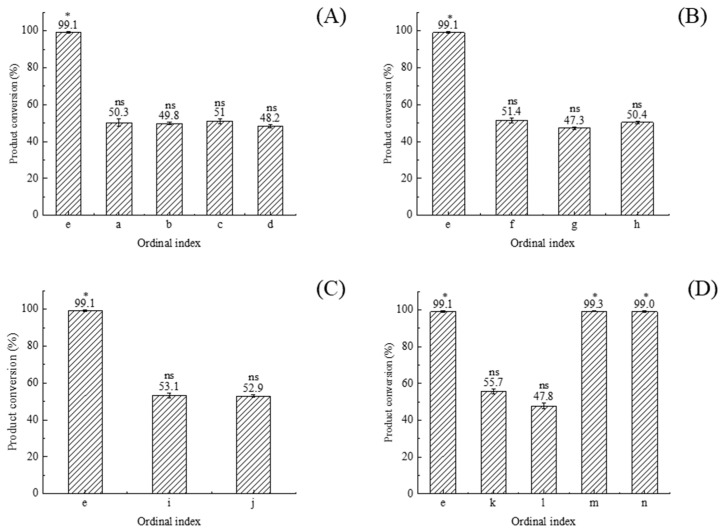
Effect of amoxicillin conversion under different conditions. *: significance of difference. ns: no significance for *p* < 0.001.((**A**): substrate concentration; (**B**): enzyme dosage; (**C**): rotational speed; (**D**): reaction temperature).

**Figure 4 ijms-25-08557-f004:**
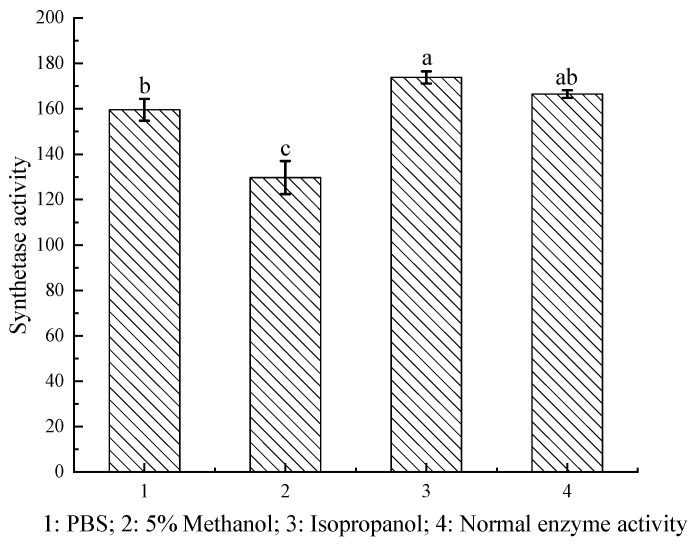
Enzyme activity after cleaning was conducted with different solvents (*p* < 0.001). (abc stands for significant difference parameter).

**Table 1 ijms-25-08557-t001:** Key parameters of the congestion effect.

	A	6-APA (g)	D-HPGM (g)	B	C	D	E	F	G	Result
a	2:1	9.08	7.61	2.09	600	20	345.00	243.95	50.3	+
b	1.5:1	6.499	5.71	2.09	600	20	345.00	248.23	49.8	+
c	1.1:1	4.76	4.19	2.09	600	20	345.00	255.79	51	+
d	1:1	4.34	3.81	2.09	600	20	345.00	258.75	48.2	+
e	1:1	4.34	3.81	4.34	600	20	166.47	166.47	99.1	−
f	1:2	4.34	3.81	4.34	600	20	345.00	256.44	51.4	+
g	1:1	4.34	3.81	3.95	600	20	182.60	139.45	47.3	+
h	1:1	4.34	3.81	4.17	600	20	173.25	129.66	50.5	+
i	1:2	4.34	3.81	4.34	800	20	345.00	252.75	53.1	+
j	1:2	4.34	3.81	4.34	1000	20	345.00	251.58	52.9	+
k	1:1.3	4.34	3.81	4.34	600	15	219.45	196.35	55.7	+
l	1:1.26	4.34	3.81	4.34	600	10	208.45	169.55	47.8	+
m	1:1.05	4.34	3.81	4.34	600	15	173.58	168.05	99.3	−
n	1:1.05	4.34	3.81	4.34	600	10	173.58	167.80	98.8	−

A: Ratio of 6-APA to D-HPGM substrate concentrations (g); B: enzyme dosage (g); C: speed of agitator (rpm); D: reaction temperature (°C); E: initial synthetase activity (U/g); F: enzymatic activity after the reaction (U/g); G: product conversion percentage (%); “+”: plugged; “−”: unplugged.

**Table 2 ijms-25-08557-t002:** Cleaning using different solvents after enzyme activity inhibition.

Solvent	Synthetase Activity 1/U/g	Synthetase Activity 2/U/g	Synthetase Activity 3/U/g	Average Synthetase Activity/U/g
PBS buffer solution	154.28	163.75	160.64	159.56
5% Methanol	124.28	126.79	138.06	129.71
5% Isopropanol	174.02	176.37	171.00	173.80
Normal enzyme activity	165.98	168.33	165.10	166.47

## Data Availability

Data is contained within the article.

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
