# Peer review of "Investigation of Enzyme Immobilization and Clogging Mechanisms in the Enzymatic Synthesis of Amoxicillin"

_ijms, 2024, doi:10.3390/ijms25168557_

Round 1

Reviewer 1 Report

Comments and Suggestions for Authors

The manuscript describes the investigation of the blocking mechanism of immobilized penicillin G acylase (PGA) in the enzymatic synthesis of amoxicillin and its solution and subsequent solution towards achieving high efficiency and sustainability in industrial production.

The topic fits the special issue “Advances in Biofuels and Green Catalysts,” the scope is adequate and the methods are appropriate.

Overall, the manuscript lacks sufficient scientific rigor and logic dissemination, with compromised reproducibility due to missing some key operating parameters (SEM, etc.).

Thus it can be accepted for publication in International Journal of Molecular Sciences after major revision.

> 10 “significant(ly)” have been used without proper statistical analysis of variance.

Equations should be numbered with proper references.

Line 154-157 how are these % values quantified? These are not reflected in Fig. 2. It is also not clear what kind of result is presented in Fig. 2. Fig. 1 does not have scale bars. Fig. 3 lacks y-axis label.  

Many statements are speculative without solid such as Line 161-168. Discussion is not in-depth, not referring to results, not evidenced by solid results (not evidence-based) with

Conclusion also lacks quantified results, if improved, by what %? Overall, the study is short of comparative measures, analysis of variance, etc. to support the claims.

Further, text editing is needed, Section 2.1 is not proper writing and does not have all the elements of a sentence, such as subject, verb. object, etc. Line 88-95 is not proper scientific writing. Should not be a manual with a series of commands.

Last, the authors have appropriately highlighted the importance and widespread efficacy of B-lactam based antibiotics. The work would be advantaged by further highlighting the fundamental chemical aspects including the strain induced by the 4-membered lactam ring, leading to 'auto-catalytic' properties of such systems. This is quite important and relevant due to the structural and synthetic nature of the present work focused on such aspects. Perhaps in the initial introductory paragraph this can be highlighted as follows:

"Due to their great efficacy and antibacterial range, related to fundamental chemical structuring and auto-catalytic properties, B-lactam antibiotics have gained widespread favor as an essential antibacterial medication in clinical treatment (Mucsi Z., 2013)." Mucsi et. al., Phys. Chem. Chem. Phys, 15 (2013), 20447-20455.

Comments on the Quality of English Language

moderate editing

Reviewer 2 Report

Comments and Suggestions for Authors

The manuscript addresses the topic of amoxicillin synthesis by immobilized penicillin G acylase (PGA), focusing on the clogging on the surface of the biocatalyst through crystallization of the substrate and the product and its reduction by removing the clogs with an organic solvent. The authors claim that their results could help achieving higher productivity and sustainability in the industrial production of amoxicillin. Although the topic of enzymatic manufacturing of β-lactam antibiotics is of scientific interest, I consider that the manuscript has serious weaknesses; therefore, I cannot recommend it for publication in a journal of this level. My main observations are listed below.

1. The manufacturing of ampicillin and other β-lactam antibiotics is a process industrialized more than 20 years ago, being one of the great achievements of industrial biocatalysis. As stated in the book of Liese, Seelbach and Wandrey the industrial repetitive batch process at the DSM company uses PGA from Escherichia coli with >90% yield and >95% selectivity, having a capacity of 2000 tons/year (Industrial Biotransformations, 2nd Ed., John Wiley-VCH, 2006, ISBN: 3-527-31001-0, pp. 401-402). Although the authors are reporting in this manuscript conversions around 99% in a single experiment, it is far from any industrial scaling-up possibility, which is not mentioned.

2. The Introduction part must be consistently revised and upgraded, to give a real overview of the state-of-the art of the research already accomplished for efficient immobilization of PGA and its utilization for the synthesis of  β-lactam antibiotics directed to industrial application. I would like to recommend in this context to read the mini-review of Li et al. (https://doi.org/10.1002/pat.4791), but a lot of other reports are available as well.

3. The experimental study of amoxicillin synthesis consists from only 14 experiments concerning the influence of the main reaction parameters: substrate concentration, enzyme dosage, rotational speed, and temperature on the synthesis of amoxicillin using an immobilized PGA which is not properly defined (mentioning the source of the enzyme, the characteristics of the support, the immobilization method, etc.) and is not compared to other similar reports from the literature, although this reaction was thoroughly investigated. Obviously, the possible blockage of the surface of the immobilized PGA (as shown by the SEM images) is closely related to the surface properties of the carrier and the bonding of the enzyme to the support; therefore, they should have been discussed in this context. Thus, the novelty and the contribution to the advancement of science in this topic are low.

4. The main contribution claimed by the authors is the identification of the clogging mechanism as caused by the crystallization of the substrate and the product. It is also mentioned that until nor “in-depth research and comprehensive reporting are still insufficient”. I cannot agree, because efficient industrial exploitation of a repetitive process is not possible without an efficient downstream solution. The recovery of the enzymatic activity is proposed here by simple washing with isopropyl alcohol. However, the processes of crystallization, mass transfer, and diffusion are more complex, as they were revealed by several other authors, like as Salami et al. (https://doi.org/10.1039/D0RE00276C).  I consider that a more comprehensive investigation is needed.

5. The legend of Figure 2 is improper.

6. Table 2. PBS buffer is not an organic solvent. “5% Methy” must be probably 5% methanol.

7. The HPLC instrument used is not specified in section 3.1. Instruments; nor the C18 column producer in lines 180-181. At lines 223-224, section 3.4.1.2. HPLC measuring must be deleted, or the whole text from lines 180-198 moved here.

8. The Conclusion part must be rewritten and focused on the real conclusions of the own work, instead of general considerations  in lines250-268.

Round 2

Reviewer 1 Report

Comments and Suggestions for Authors

The majority of the issues have been addressed.

However, the following 2 remain to be address:

1. Section 3.1, the same grammar problem remains.

2. Although p values have been reported, the statistical analysis applied has not been introduced in the Methods.

3. Conclusion is overly long, please refine.

Comments on the Quality of English Language

minor editing

Reviewer 2 Report

Comments and Suggestions for Authors

As concerns my observations and recommendations regarding the original manuscript, I noticed that the point-to-point answers of the authors were mostly pertinent and the manuscript was consistently revised in accordance to them, being significantly improved compared to the original one. I can also accept that the opinions of authors and reviewers could be different, but I still consider that at least the inclusion of a study regarding the repeated use of the biocatalyst (in 10 reaction cycles or even more) would have been important to demonstrate the efficiency of the proposed clogging method and its suitability for a possible scaling-up. I hope that such research will be scheduled in the next future.
